# Mathematical Characterization of Better-than-Random Multiclass Models

**Sébastien Foulle**  *sebastien.foulle@abeille-assurances.fr*
*Marketing, Customer Experience and Institutional Relations Department*
*Abeille Assurances*
*80 Avenue de l'Europe*
*92270 Bois-Colombes, France*

**Reviewed on OpenReview:** *https: // openreview. net/ forum? id= VdW9SkALSd*

## Abstract

A binary supervised model outperforms chance if and only if the determinant of the confusion matrix is positive. This is equivalent to saying that the associated point in the ROC space is above the random guessing line. This also means that Youden's J, Cohen's $\kappa$ and Matthews' correlation coefficient are positive. We extend these results to any number of classes: for a target variable with $m \geq 2$ classes, we show that a model does better than chance if and only if the entries of the confusion matrix verify $m(m-1)$ homogeneous polynomial inequalities of degree 2, which can be expressed using generalized likelihood ratios. We also obtain a more theoretical formulation: a model does better than chance if and only if it is a maximum likelihood estimator of the target variable. When this is the case, we find that the multiclass versions of the previous metrics remain positive. If $m > 2$, we notice that no-skill classifiers are only a small part of the topological boundary between better-than-random models and bad models. For $m = 3$, we show that bad models occupy exactly 90% of the ROC space, far more than the 50% of the two-class problems. Finally, we propose to define weak multiclass classifiers by conditions on these generalized likelihood ratios.

## 1 Introduction

In machine learning, supervised multiclass modeling seeks to predict as correctly as possible the values of a categorical target variable that takes at least $m \geq 3$ distinct values. The necessary and sufficient condition for correct binary modeling is well known: we must have sensitivity $\geq 1 -$ specificity, equality corresponding to random models. The ROC curve of a probabilistic model must therefore lie above the line segment of no-skill classifiers, and the area under the ROC curve is between that of the perfect model and the area under this line.

The conditions for correct modeling have not received much attention when $m \geq 3$. Practitioners have successfully focused on designing good multiclass algorithms, with model performance measured, for example, by macro-averaging classical binary metrics. But this doesn't provide a clear understanding of the multiclass case, especially when it comes to generalizing theoretical concepts such as AUC or other metrics. From a more algorithmic perspective, ensemble learning combines weak classifiers, that is models that perform slightly better than random guessing. Weak multiclass classifiers have been built by avoiding, for example, conditions on accuracy that are too strong or too weak (Schapire & Freund 2012, Chapter 10). Knowing the exact conditions to be satisfied can thus help in the design of these algorithms. Our main objective is to obtain a mathematical characterization of multiclass models that do better than chance, which we call *decent models*. However, it is beyond the scope of this article to study cost-sensitive models, multiclass probabilistic classification or which multiclass metric is best suited to a given use case, be it Matthews' correlation coefficient, Cohen's $\kappa$ or the volume under the ROC surface, for example.

Researchers have concentrated on the actual construction of the ROC hypersurface and the calculation of the associated volume (see Kleiman & Page 2019), which do not require defining what a better-than-random model is. The only notable exception is in Ferri et al. (2003), which characterizes these models when $m = 3$. Edwards et al. (2005) state a second theoretical result: for any $m > 2$, and with a certain choice of axes in the ROC space, the hypervolumes under the ROC hypersurface of a perfect model ("perfect ideal observer") and a no-skill classifier ("guessing observer") are equal to 0. The authors therefore observe that we no longer have a continuous variation of the AUC when the difficulty of the task to be modelled varies from the simplest to the most complex, unlike binary classification. And Powers (2011) proposes multiclass generalizations of several binary metrics, for example in the form of fractions with a positive denominator and as numerator the determinant of the confusion matrix. In what follows, we show that some of these results are incorrect, that others are confusing, and that the definitions of some multiclass metrics are unconvincing.

To get to the multiclass setting, we need concepts that are simpler to state and understand when $m = 2$. The first section recalls the definitions of classical metrics as well as those of lifts, which are metrics mainly used in marketing. We then describe how to characterize decent binary models using these different metrics.

The second section formalizes a well-known property of certain binary metrics, the *prevalence-invariance*. These metrics do not depend on prevalence and therefore measure model characteristics. We establish links between lifts, which depend on prevalence, and positive and negative likelihood ratios, which are prevalence-invariant. These links are decisive in achieving our main result.

The third section is devoted to the new results we derive for the multiclass case. We first characterize random models and generalize some of the metrics seen in the two-class tasks. We relate these metrics to binary One versus Rest models, and we state the theorem that links lifts and generalized likelihood ratios. This result makes it possible to define decent (or random) multiclass models as follows: all likelihood ratios must be greater than or equal to one. It is easy to reformulate these conditions for any $m \geq 2$: a model does better than chance if and only if it is a maximum likelihood estimator of the target variable. The properties of these models are illustrated on different examples. In particular, we show that a bad model can look good on one data set and bad on the over- or undersampled data set. This phenomenon does not appear to have been noted before and only occurs for $m > 2$. We then check that multiclass versions of Cohen's $\kappa$, Matthews' correlation coefficient and a generalization of Youden's J take positive values on decent models. And we prove that the multiclass balanced accuracy of a decent model is always greater than $\frac{1}{m}$, this bound being optimal. We also establish that our multiclass version of Youden's J satisfies a property not verified by an analogous formula proposed by Powers (2011). In the next section, we characterize the topological boundary between decent and bad models. In particular, we note that random models are only a small subset of this boundary, which leads us to qualify Edwards et al. (2005)'s conclusions. We then examine an earlier criterion in Ferri et al. (2003) and show that it is too restrictive in its characterization of bad three-class models. This leads us to recalculate the volume occupied by bad three-class models in the six-dimensional space of all three-class models. This volume represents 90% of the total volume, much more than the 50% of the ROC space when $m = 2$. We conclude with a proposed definition of weak multiclass models based on generalized likelihood ratios. In particular, we derive minimum and maximum values for the multiclass balanced accuracy of a weak classifier.

## 2 Reminders on Binary Models and Metrics

We review how to characterize binary models that do better than chance, using several metrics that we'll find again in the multiclass case.

**Definition 1.**
- *A binary target variable is a function $y : \mathcal{O} \to \{0, 1\}$ defined on a set of observations, and the class of interest is $y = 1$. A binary predictive model of $y$ is a function $\hat{y} : \mathcal{O} \to \{0, 1\}$ obtained by a supervised learning algorithm.*

- *The confusion matrix $\mathrm{mat}(a, b, c, d)$ counts the crossed values of $y$ and $\hat{y}$:*

|  | $\hat{y} = 0$ | $\hat{y} = 1$ |
|---|---|---|
| $y = 0$ | $a$ | $b$ |
| $y = 1$ | $c$ | $d$ |

- *The number of observations is $n = a + b + c + d$.*

- *The prevalence is $\lambda = \mathbb{P}(y = 1) = \frac{c+d}{n}$. We assume that $0 < \lambda < 1$, that is $y(\mathcal{O}) = \{0, 1\}$ (in other words, each row of the confusion matrix contains at least one non-zero entry).*

- *The detection prevalence (or positivity rate) is $\mu = \mathbb{P}(\hat{y} = 1) = \frac{b+d}{n}$.*

### 2.1 Classical Metrics

We recall the definitions of several well-known metrics. For two events $A$ and $B$, $\mathbb{P}(A \mid B) = \frac{\mathbb{P}(A \cap B)}{\mathbb{P}(B)}$ is the conditional probability of $A$ knowing $B$ and $A^{\mathsf{c}} = \mathcal{O} - A$ is the complementary of $A$.

**Definition 2.**

$$\text{sensitivity} = \mathbb{P}(\hat{y} = 1 \mid y = 1) = \frac{d}{c+d} = \frac{d/n}{\lambda}$$

$$\text{specificity} = \mathbb{P}(\hat{y} = 0 \mid y = 0) = \frac{a}{a+b} = \frac{a/n}{1 - \lambda}$$

$$\text{precision} = \mathbb{P}(y = 1 \mid \hat{y} = 1) = \frac{d}{b+d} = \frac{d/n}{\mu}$$

$$\text{precision}_0 = \mathbb{P}(y = 0 \mid \hat{y} = 0) = \frac{a}{a+c} = \frac{a/n}{1 - \mu}$$

**Youden's index** $J = \text{sensitivity} + \text{specificity} - 1 = \dfrac{(ad - bc)/n^2}{\lambda(1 - \lambda)}$

**Matthews' correlation coefficient** $\text{MCC} = \text{corr}(y, \hat{y}) = \dfrac{(ad - bc)/n^2}{\sqrt{\lambda(1 - \lambda)\mu(1 - \mu)}}$

**Cohen's** $\kappa = \dfrac{\mathbb{P}(y = \hat{y}) - \lambda\mu - (1 - \lambda)(1 - \mu)}{\lambda(1 - \mu) + \mu(1 - \lambda)} = \dfrac{2(ad - bc)/n^2}{\lambda(1 - \mu) + \mu(1 - \lambda)}$

**Positive likelihood ratio** $\text{LR}_+ = \dfrac{\text{sensitivity}}{1 - \text{specificity}} = \dfrac{d(1 - \lambda)}{b\lambda}$

**Negative likelihood ratio** $\text{LR}_- = \dfrac{1 - \text{sensitivity}}{\text{specificity}} = \dfrac{c(1 - \lambda)}{a\lambda}$

**Diagnostic odds ratio** $\text{DOR} = \dfrac{\text{LR}_+}{\text{LR}_-} = \dfrac{ad}{bc}$

**Relative risk** $\text{RR} = \dfrac{\text{precision}}{1 - \text{precision}_0} = \dfrac{d(1 - \mu)}{c\mu}$

The definition of $\kappa$ requires some explanation: we start with the probability $\mathbb{P}(y = \hat{y}) = \mathbb{P}(y = \hat{y} = 1) + \mathbb{P}(y = \hat{y} = 0) = \frac{a+d}{n}$. Then we subtract this same expression when $y$ and $\hat{y}$ are independent, which is $\mathbb{P}(y = 1)\mathbb{P}(\hat{y} = 1) + \mathbb{P}(y = 0)\mathbb{P}(\hat{y} = 0) = \lambda\mu + (1 - \lambda)(1 - \mu)$. And we divide by 1 minus this expression to normalize the result. The denominator is therefore $1 - \lambda\mu - (1 - \lambda)(1 - \mu) = \lambda + \mu - 2\lambda\mu = \lambda(1 - \mu) + \mu(1 - \lambda)$.

Finally, we recall the definitions of the ROC curve (or Receiver Operating Characteristic curve) and the ROC space (see for example Davis & Goadrich 2006). The ROC space is the square $[0, 1] \times [0, 1]$ in which each binary model is represented by the point with coordinates $(1 - \text{specificity}, \text{sensitivity})$. It is made up of three distinct parts: the straight line with equation $J = 0$ corresponds to random models. The upper triangular region defined by $J > 0$ corresponds to models that do better than chance, also called "decent models" in the following. And the lower triangular region defined by $J < 0$ corresponds to models that do worse than chance. If $p : \mathcal{O} \to \mathbb{R}$ is a predictive score of $y$, for any threshold $t \in \mathbb{R}$ we define a binary model $\hat{y}_t$ by $\hat{y}_t = 1$ if $p \geq t$ and $\hat{y}_t = 0$ otherwise. In ROC space, the ROC curve connects points $\hat{y}_t$ and AUC stands for Area Under the ROC Curve.

### 2.2 Lifts

In addition to the above, we will need the lift (see Berrar, 2019) so here's a reminder of its main characteristics.

**Definition 3.** *For two events $A$ and $B$ with non-zero probability, lift is defined by*

$$\text{Lift}(A, B) = \frac{\mathbb{P}(A \cap B)}{\mathbb{P}(A)\mathbb{P}(B)} = \frac{\mathbb{P}(A \mid B)}{\mathbb{P}(A)} = \frac{\mathbb{P}(B \mid A)}{\mathbb{P}(B)}$$

*Events $A$ and $B$ are said to attract (resp. are independent, repel) if $\text{Lift}(A, B) > 1$ (resp. $= 1$, $< 1$). And for any model $\hat{y}$ of a target variable $y$ such that $\mu > 0$, we note $\text{lift} = \text{Lift}(y = 1, \hat{y} = 1)$ for short.*

To say that $A$ and $B$ attract is equivalent to saying that we have $\mathbb{P}(A \mid B) > \mathbb{P}(A)$ and $\mathbb{P}(B \mid A) > \mathbb{P}(B)$. In other words, if one of the two events occurs, the probability of observing the other event increases.

Lift has the following properties:

- Lift is symmetric: $\text{Lift}(A, B) = \text{Lift}(B, A)$.

- Lift is easily interpreted: "model $\hat{y}$ of $y$ detects the positive class *lift* times better than chance", because $\mathbb{P}(y = 1 \mid \hat{y} = 1) = \text{lift}\,\mathbb{P}(y = 1)$.

- Lemma 1 shows that $\text{Lift}(A, B) - 1$ and $\text{Lift}(A^c, B) - 1$ have opposite signs.

**Lemma 1.** *For two events $A$ and $B$ with $0 < \mathbb{P}(A) < 1$ and $0 < \mathbb{P}(B)$, we have*

$$\mathbb{P}(A)\left(\text{Lift}(A, B) - 1\right) = \mathbb{P}(A^c)\left(1 - \text{Lift}(A^c, B)\right)$$

*So for $0 < \lambda < 1$ and $0 < \mu < 1$*

$$\text{Lift}(y = 1, \hat{y} = 1) - 1 = \frac{1 - \lambda}{\lambda}(1 - \text{Lift}(y = 0, \hat{y} = 1))$$

$$= \frac{1 - \mu}{\mu}(1 - \text{Lift}(y = 1, \hat{y} = 0))$$

$$= \frac{(1 - \lambda)(1 - \mu)}{\lambda\mu}(\text{Lift}(y = 0, \hat{y} = 0) - 1)$$

*Proof.* By definition

$$\mathbb{P}(A)\left(\text{Lift}(A, B) - 1\right) = \mathbb{P}(A)\left(\frac{\mathbb{P}(A \cap B)}{\mathbb{P}(A)\mathbb{P}(B)} - 1\right) = \mathbb{P}(A \mid B) - \mathbb{P}(A)$$

Replacing $A$ by $A^c$ gives $\mathbb{P}(A^c)\left(\text{Lift}(A^c, B) - 1\right) = \mathbb{P}(A^c \mid B) - \mathbb{P}(A^c)$. Then $\mathbb{P}(A \mid B) - \mathbb{P}(A) = 1 - \mathbb{P}(A^c \mid B) - \mathbb{P}(A) = \mathbb{P}(A^c) - \mathbb{P}(A^c \mid B)$, giving $\mathbb{P}(A)\left(\text{Lift}(A, B) - 1\right) = -\mathbb{P}(A^c)\left(\text{Lift}(A^c, B) - 1\right) = \mathbb{P}A^c)\left(1 - \text{Lift}(A^c, B)\right)$ □

We can have $\mu = 0$ or $\mu = 1$ when $\hat{y}$ is a constant model equal to 0 or 1. If $\mu = 0$, $\text{Lift}(y = 1, \hat{y} = 1)$ and $\text{Lift}(y = 0, \hat{y} = 1)$ do not exist. If $\mu = 1$, $\text{Lift}(y = 1, \hat{y} = 0)$ and $\text{Lift}(y = 0, \hat{y} = 0)$ do not exist.

**Proposition 1.** *We have the following equalities (when all members are well defined):*

$$\text{Lift}(y = 1, \hat{y} = 1) - 1 = \frac{(ad - bc)/n^2}{\lambda\mu}$$

$$1 - \text{Lift}(y = 0, \hat{y} = 1) = \frac{(ad - bc)/n^2}{(1 - \lambda)\mu}$$

$$\text{Lift}(y = 0, \hat{y} = 0) - 1 = \frac{(ad - bc)/n^2}{(1 - \lambda)(1 - \mu)}$$

$$1 - \text{Lift}(y = 1, \hat{y} = 0) = \frac{(ad - bc)/n^2}{\lambda(1 - \mu)}$$

*Proof.* By definition, $\lambda\mu(\text{lift} - 1) = \mathbb{P}(y = \hat{y} = 1) - \lambda\mu = \frac{d(a + b + c + d) - (b + d)(c + d)}{n^2} = \frac{ad - bc}{n^2}$. From Lemma 1, we can deduce the formulas for the other lifts, assuming $\mu(1 - \mu)$ to be non-zero. By positing $\mu = 0$ then $\mu = 1$, we can check that the formulas remain true (when both members of the equation are well defined). □

### 2.3 Decent Binary Models

Models fall into three categories:

- Those that do better than chance.

- Models that work like chance.

- Models that do worse than chance.

For a model of the first category, we want the events $\{y = 1\}$ and $\{\hat{y} = 1\}$ to attract, and similarly for the events $\{y = 0\}$ and $\{\hat{y} = 0\}$. Proposition 1 shows that this is equivalent to $ad - bc > 0$. Models that proceed randomly are those that are independent of the target variable, which is equivalent to saying that all lifts are equal to 1, that is $ad - bc = 0$. The constants 0 and 1 are part of these models, and constants are not random phenomena in the usual sense of the term. We therefore prefer to refer to them as uninformative rather than random models. The other models are characterized by $ad - bc < 0$. For these, the events $\{y = 1\}$ and $\{\hat{y} = 0\}$ attract each other, as well as $\{y = 0\}$ and $\{\hat{y} = 1\}$. In other words, these models tend to predict 0 instead of 1 and vice versa. This is most often the result of an accidental permutation of the class labels. This leads us to the next definition.

**Definition 4.**
- *A model is* decent *if $ad - bc > 0$.*

- *A model is* uninformative *if $ad - bc = 0$.*

- *A model is* erroneous *if $ad - bc < 0$.*

An erroneous model can be transformed into a decent model by replacing $\hat{y}$ by $1 - \hat{y}$ (we will find out on an example that in the multiclass setting, swapping the values taken by $\hat{y}$ doesn't transform every bad model into a decent one). And an uninformative model $\hat{y}$ is a model characterized by the value $\mu = \mathbb{P}(\hat{y} = 1)$. It has sensitivity $\mathbb{P}(\hat{y} = 1 \mid y = 1) = \mathbb{P}(\hat{y} = 1) = \mu$ and specificity $\mathbb{P}(\hat{y} = 0 \mid y = 0) = \mathbb{P}(\hat{y} = 0) = 1 - \mu$.

The next proposition shows that the category of a binary model can be deduced from the values taken by any of the usual metrics. We'll see later that a single metric is no longer sufficient in the multiclass case.

**Proposition 2.** *The following statements are equivalent:*

- *Model $\hat{y}$ of $y$ is decent*

- $\text{Lift}(y = 1, \hat{y} = 1) > 1$

- $\text{Lift}(y = 0, \hat{y} = 0) > 1$

- $\text{Lift}(y = 1, \hat{y} = 0) < 1$

- $\text{Lift}(y = 0, \hat{y} = 1) < 1$

- $J > 0$

- $\text{MCC} > 0$

- $\kappa > 0$

- $\text{LR}_+ > 1$

- $\text{LR}_- < 1$

- $\text{DOR} > 1$

*If any of these inequalities is an equality, the model is uninformative and all inequalities are equalities.*

*Proof.* This is obvious for the four lifts and metrics $J$, MCC, $\kappa$ and DOR from Definition 2 and Proposition 1. Using $J$, a model is decent (resp. uninformative) if and only if sensitivity $> 1 -$ specificity (resp. $= 1 -$ specificity) that is specificity $> 1 -$ sensitivity (resp. $= 1 -$ sensitivity). By division we deduce the desired result for $\text{LR}_+$ and $\text{LR}_-$. □

There are two other ways of interpreting the three categories of models, based on the sign of $J$ or MCC:

- In the ROC space, a decent (resp. uninformative, erroneous) model is above (resp. on, below) the diagonal line with equation sensitivity $= 1 -$ specificity.

- We have models that are positively correlated with the target variable, those that are independent of it, and finally models that are negatively correlated with the target variable.

## 3 Links between Prevalence and Metrics

Suppose we have a binary target variable $y : \mathcal{O} \to \{0, 1\}$. If we train a model on $\mathcal{O}$ and apply it to a test data set $\mathcal{T}$, the statistical distribution of the characteristics of observations verifying $y = 1$ (resp. $y = 0$) must be approximately the same on $\mathcal{O}$ and $\mathcal{T}$. On the other hand, the prevalence of the test data set may differ from that of the training data set: we sometimes over- or under-sample the data to train the model. In practice, therefore, a model is applied to different data sets, which differ from each other only in their prevalence (apart from random variations). So a model $\hat{y}$ applied to two data sets will have approximately confusion matrices $\text{mat}(a, b, c, d)$ and $\text{mat}(ua, ub, vc, vd)$ with $u > 0$ and $v > 0$.

### 3.1 Prevalence-Invariance

We formalize a well-known property here, and obtain a first new result: Youden's $J$ allows us to characterize models with poor predictive ability, whatever the values of $\lambda$ and $\mu$.

**Definition 5.** *A metric is* prevalence-invariant *(also noted $\lambda$-invariant) if it does not depend on prevalence. In other words, the metric applied to the confusion matrix below must not depend on the positive numbers $u$ and $v$.*

|  | $\hat{y} = 0$ | $\hat{y} = 1$ |
|---|---|---|
| $y = 0$ | ua | ub |
| $y = 1$ | vc | vd |

This is equivalent to saying that such a metric is a function of $\frac{b}{a}$ and $\frac{c}{d}$ (by posing $u = \frac{1}{a}$ and $v = \frac{1}{d}$). As specificity $= \frac{1}{1+\frac{b}{a}}$ and sensitivity $= \frac{1}{1+\frac{c}{d}}$, $\lambda$-invariant metrics are functions of sensitivity and specificity. Among the metrics we've already encountered, $J$, $\text{LR}_+$, $\text{LR}_-$ and DOR are well known for being $\lambda$-invariant.

We note that the sign of $ad - bc$ does not change if we multiply the rows of the matrix by $u > 0$ and $v > 0$. In other words, whether a model is decent, uninformative or erroneous does not depend on prevalence. In the multiclass case, we'll see that this property isn't automatic and that it's an integral part of the definition of a decent model. In general, $\lambda$-invariant metrics can be used to measure the absolute quality of a model, as shown in Example 1.

**Example 1.** *Consider a first data set $D$, a binary target variable $y$ and a model $\hat{y}$ of $y$ described by the following confusion matrix. We claim that this model is excellent, albeit imperfect.*

|  | $\hat{y} = 0$ | $\hat{y} = 1$ |
|---|---|---|
| $y = 0$ | 999 | 1 |
| $y = 1$ | 1 | 999 |

*We now consider a second data set $E$, obtained from the first by replicating the negative class millions of times. The model $\hat{y}$ applied to this second data set has the following confusion matrix. Model precision is equal to 0.999 on $D$ and 0.000998 on $E$.*

|  | $\hat{y} = 0$ | $\hat{y} = 1$ |
|---|---|---|
| $y = 0$ | 999 000 000 | 1 000 000 |
| $y = 1$ | 1 | 999 |

Does this mean that a model can be both good and bad? Obviously not. But if we use a non $\lambda$-invariant metric, even the best model will look bad on a very unbalanced data set. On the other hand, with a $\lambda$-invariant metric, the model quality estimate is independent of the prevalence of the data set used. It is therefore a model-specific quality that is measured.

Suppose we want to detect poorly predictive models. This is a property of models, so only a $\lambda$-invariant metric can reveal it. The metric J fits the bill perfectly, as the next proposition shows.

**Proposition 3.** *Let* $K = \max(|\text{sensitivity} - \mu|, |\text{specificity} - 1 + \mu|)$. *We have*

$$0 \leq K \leq |J| \leq 2K$$

*Proof.* Given $\phi = \text{sensitivity} - \mu$ and $\psi = \text{specificity} - 1 + \mu$, we have $\phi = \mathbb{P}(\hat{y} = 1 \mid y = 1) - \mu = \frac{\mathbb{P}(y = \hat{y} = 1)}{\lambda} - \mu = \mu \, \text{lift} - \mu = \frac{(ad - bc)/n^2}{\lambda} = (1 - \lambda)J$ according to Proposition 1. We deduce $\psi = J - \text{sensitivity} + \mu = J - (1 - \lambda)J = \lambda J$, hence $0 \leq K \leq |J|$. Furthermore $J = \phi + \psi$ and these three quantities have the same sign, that of $ad - bc$. So we have $|J| = |\phi| + |\psi|$ which gives $|J| \leq 2K$. $\square$

According to this proposition, $J$ is small if and only if the sensitivity and specificity of the model are close to their values for uninformative models. This result had been observed numerically in Chicco et al. (2021).

### 3.2 Relationships between Likelihood Ratios and Lifts

Likelihood ratios are also $\lambda$-invariant metrics, and the next proposition shows their narrow links with lifts (see Vu et al. 2019 for similar formulas): we can express likelihood ratios as functions of lifts and vice versa. These formulas will be used to define generalized likelihood ratios and generalized diagnostic odds ratios in the multiclass setting.

**Proposition 4.** *We have the following equations:*

$$\text{LR}_+ = \frac{\text{Lift}(y = 1, \hat{y} = 1)}{\text{Lift}(y = 0, \hat{y} = 1)}$$

$$\text{LR}_- = \frac{\text{Lift}(y = 1, \hat{y} = 0)}{\text{Lift}(y = 0, \hat{y} = 0)}$$

$$\text{DOR} = \frac{\text{Lift}(y = 1, \hat{y} = 1)\text{Lift}(y = 0, \hat{y} = 0)}{\text{Lift}(y = 1, \hat{y} = 0)\text{Lift}(y = 0, \hat{y} = 1)}$$

$$\text{lift} - 1 = (1 - \lambda)\frac{\text{LR}_+ - 1}{\lambda(\text{LR}_+ - 1) + 1}$$

$$1 - \text{Lift}(y = 1, \hat{y} = 0) = (1 - \lambda)\frac{1 - \text{LR}_-}{1 - \lambda(1 - \text{LR}_-)}$$

*Proof.* By definition $\text{Lift}(y = 1, \hat{y} = 1) = \frac{d/n}{\lambda\mu}$, $\text{Lift}(y = 0, \hat{y} = 1) = \frac{b/n}{(1 - \lambda)\mu}$, $\text{Lift}(y = 1, \hat{y} = 0) = \frac{c/n}{\lambda(1 - \mu)}$ and $\text{Lift}(y = 0, \hat{y} = 0) = \frac{a/n}{(1 - \lambda)(1 - \mu)}$. We then obtain the first three equalities by applying Definition 2. We also have $\mu = \mathbb{P}(\hat{y} = 1) = \mathbb{P}(\{\hat{y} = 1\} \cap \{y = 1\}) + \mathbb{P}(\{\hat{y} = 1\} \cap \{y = 0\}) = \mathbb{P}(y = 1)\mathbb{P}(\hat{y} = 1 \mid y = 1) + \mathbb{P}(y = 0)\mathbb{P}(\hat{y} = 1 \mid y = 0) = \lambda \, \text{sensitivity} + (1 - \lambda)(1 - \text{specificity})$. This results in

$$\text{lift} - 1 = \frac{\text{sensitivity}}{\mu} - 1 = \frac{\text{sensitivity}}{\lambda \, \text{sensitivity} + (1 - \lambda)(1 - \text{specificity})} - 1$$

$$= \frac{\text{LR}_+}{\lambda\text{LR}_+ + 1 - \lambda} - 1 = \frac{\text{LR}_+ - \lambda(\text{LR}_+ - 1) - 1}{\lambda(\text{LR}_+ - 1) + 1} = \frac{(1 - \lambda)(\text{LR}_+ - 1)}{\lambda(\text{LR}_+ - 1) + 1}$$

We deduce the second equation by exchanging the values 0 and 1 of $\hat{y}$. $\qquad\square$

We can make a metric $\lambda$-invariant by making $\lambda$ tend towards 0.

**Corollary 1.** *If sensitivity and specificity are fixed:*

$$\mathrm{LR}_+ = \lim_{\lambda \to 0} \mathrm{lift}$$

$$\mathrm{LR}_- = \lim_{\lambda \to 0} \mathrm{Lift}(y = 1, \hat{y} = 0)$$

$$\mathrm{DOR} = \lim_{\lambda \to 0} \frac{\mathrm{Lift}(y = 1, \hat{y} = 1)}{\mathrm{Lift}(y = 1, \hat{y} = 0)} = \lim_{\lambda \to 0} \mathrm{RR}$$

*Proof.* We make $\lambda$ tend towards 0 in the last two equations of Proposition 4 to obtain the first two results. We then apply the formulas $\mathrm{DOR} = \frac{\mathrm{LR}_+}{\mathrm{LR}_-}$ and $\mathrm{RR} = \frac{d(1-\mu)}{c\mu} = \frac{\mathrm{Lift}(y=1,\hat{y}=1)}{\mathrm{Lift}(y=1,\hat{y}=0)}$. $\qquad\square$

### 3.3 Normalized Likelihood Ratios

This section is not useful for preparing the definition of decent multiclass models. But the three new metrics defined here may be of interest to practitioners of likelihood ratios and odds ratios : by applying the same transformation to $\mathrm{LR}_+$, $\mathrm{LR}_-$ and $\mathrm{DOR}$, we obtain metrics with a wider definition domain, which are zero for uninformative models and take the value 1 if the model is perfect. We briefly need the following notations, with "CO" standing for "centered odds".

**Definition 6.**

$$CO(\mathrm{LR}_+) = \frac{\mathrm{LR}_+ - 1}{\mathrm{LR}_+ + 1} = \frac{J}{\mathrm{sensitivity} + 1 - \mathrm{specificity}}$$

$$CO(\mathrm{LR}_-) = \frac{1 - \mathrm{LR}_-}{1 + \mathrm{LR}_-} = \frac{J}{1 - \mathrm{sensitivity} + \mathrm{specificity}}$$

$$CO(\mathrm{DOR}) = \frac{\mathrm{DOR} - 1}{\mathrm{DOR} + 1} = \frac{ad - bc}{ad + bc}$$

Another simple way to make a metric $\lambda$-invariant is to set $\lambda = 0.5$. Using Proposition 4, we obtain

$$\mathrm{lift}|_{\lambda=0.5} - 1 = CO(\mathrm{LR}_+) = 1 - \frac{2}{\mathrm{LR}_+ + 1}$$

This metric has a larger domain of definition than $\mathrm{LR}_+$ as it is well defined as soon as $\mu \neq 0$. And it's clear that the best of a set of models for $\mathrm{LR}_+$ is also the best for this new metric. They have two similar geometric interpretations. Let $M$ be a point on the ROC curve and $O$ the origin:

- $\mathrm{LR}_+(M)$ is the tangent of the angle between the line segment $[OM]$ and the $x$-axis.

- $CO(\mathrm{LR}_+)(M)$ is the tangent of the angle between the line segment $[OM]$ and the line of equation $y = x$ because $\tan\left(\omega - \frac{\pi}{4}\right) = \frac{\tan(\omega)-1}{\tan(\omega)+1}$.

We also have $1 - \mathrm{Lift}(y = 1, \hat{y} = 0)|_{\lambda=0.5} = CO(\mathrm{LR}_-) = \frac{2}{1+\mathrm{LR}_-} - 1$. This metric is defined as soon as $\mu \neq 1$ and here, too, the best model for one of the two metrics is the best for the other. Finally, the metric $CO(\mathrm{DOR}) = 1 - \frac{2}{\mathrm{DOR}+1}$ is defined for $\mu(1 - \mu) \neq 0$ and both metrics always have the same best model. It is clear that $CO(\mathrm{LR}_+)$, $CO(\mathrm{LR}_-)$ and $CO(\mathrm{DOR})$ lie between 0 and 1 for decent models. These transformations can be interpreted in terms of odds: for any $x \geq 0$, there exists $p \in [0, 1]$ such that $x = \frac{p}{1-p}$, and then we have $\frac{x-1}{x+1} = 2p - 1 \in [-1, 1]$.

## 4 Multiclass Models and Metrics

We consider an integer $m \geq 3$, a target variable $y : \mathcal{O} \to \{0, 1, ..., m-1\}$ such that $y(\mathcal{O}) = \{0, 1, ..., m-1\}$ and a model $\hat{y} : \mathcal{O} \to \{0, 1, ..., m-1\}$. Let $n_{i,j}$ be the number of observations in the set $\{y = i\} \cap \{\hat{y} = j\}$ and $(n_{i,j})_{i,j}$ the confusion matrix. We note $n_{i.} = \sum_j n_{i,j}$, $n_{.j} = \sum_i n_{i,j}$, $n = \sum_j n_{.j} = \sum_i n_{i.}$, $\lambda_i = \frac{n_{i.}}{n}$ and $\mu_j = \frac{n_{.j}}{n}$. By definition we have $\lambda_i > 0$ for any $i$, and when $m = 2$ we get $\lambda_0 = 1 - \lambda$, $\lambda_1 = \lambda$, $\mu_0 = 1 - \mu$ and $\mu_1 = \mu$.

All the results in this section are new, apart from the next characterization of uninformative models and the multiclass definitions of Matthews' correlation coefficient and Cohen's $\kappa$.

### 4.1 Models

An uninformative model is one that tells us nothing about the target variable, that is $y$ and $\hat{y}$ are independent variables. We can rephrase this condition as follows (see the "guessing" observer in Edwards et al. 2005).

**Proposition 5.** *The model $\hat{y}$ of $y$ is uninformative if and only if the confusion matrix is of rank 1.*

*Proof.* If $\hat{y}$ is uninformative, we have $\mathbb{P}(\{y = i\} \cap \{\hat{y} = j\}) = \mathbb{P}(y = i)\mathbb{P}(\hat{y} = j)$ for any $i$ and $j$, which is equivalent to $n_{i,j}n = n_{i.}n_{.j}$ for any $i$ and $j$. All the columns of the confusion matrix are then proportional to the non-zero vector $(n_{i.})_i$ (and the rows are proportional to the vector $(n_{.j})_j$), which implies that the confusion matrix is of rank 1.

Conversely, if this matrix is of rank 1, the non-zero columns of the matrix are proportional to each other, and therefore proportional to their sum $(n_{i.})_i$. So there exists a family of numbers $(\beta_j)_j$ such that $n_{i,j} = n_{i.}\beta_j$ for any $i$ and $j$. Summing this equations over $i$ gives $n_{.j} = \beta_j n$, hence $n_{i,j}n = n_{i.}n_{.j}$ for any $i$ and $j$. $\square$

This proposition shows that the model $\hat{y}$ of $y$ is uninformative if and only if there are two families of numbers $(\alpha_i)_i$ and $(\beta_j)_j$ such that $\mathbb{P}(\{y = i\} \cap \{\hat{y} = j\}) = \alpha_i\beta_j$ for any $i$ and $j$. On a balanced data set, the confusion matrix of an uninformative model has all its rows equal, and its rate of well-classified observations is therefore $\frac{1}{m}$. We'll see later how to generalize this result to any data set, by replacing accuracy with balanced accuracy. It may be tempting to extend the metric formulas of the binary case by replacing $ad - bc$ by the determinant of the confusion matrix $(n_{i,j})_{i,j}$ (see for example Powers 2011), but this is not a good idea: this determinant is zero if and only if $(n_{i,j})_{i,j}$ is of rank $< m$, which is a much weaker condition than the one in the previous proposition. We will see later an example of a decent multiclass model with a confusion matrix whose determinant is zero.

We generalize part of Proposition 4 to define the following pointwise metrics.

**Definition 7.** *For any $i$ and $j$, let*

$$\text{Lift}(y = i, \hat{y} = j) = \frac{\mathbb{P}(\{y = i\} \cap \{\hat{y} = j\})}{\lambda_i \mu_j}$$

$$\text{LR}_{i,j} = \frac{\mathbb{P}(\hat{y} = j \mid y = j)}{\mathbb{P}(\hat{y} = j \mid y = i)} = \frac{\text{Lift}(y = j, \hat{y} = j)}{\text{Lift}(y = i, \hat{y} = j)} = \frac{n_{j,j}}{n_{i,j}}\frac{\lambda_i}{\lambda_j}$$

$$\text{DOR}_{i,j} = \text{DOR}_{j,i} = \text{LR}_{i,j}\text{LR}_{j,i} = \frac{n_{i,i}n_{j,j}}{n_{i,j}n_{j,i}} = \frac{\mathbb{P}(\hat{y} = j \mid y = j)/\mathbb{P}(\hat{y} = i \mid y = j)}{\mathbb{P}(\hat{y} = j \mid y = i)/\mathbb{P}(\hat{y} = i \mid y = i)}$$

For $m = 2$ we find $\text{LR}_{0,1} = \text{LR}_+$, $\text{LR}_{1,0} = \frac{1}{\text{LR}_-}$ and $\text{DOR}_{0,1} = \text{DOR}$. The next result generalizes Proposition 1 and shows a direct link between diagonal lifts $\text{Lift}(y = i, \hat{y} = i)$ and One versus Rest binary models.

**Definition 8.** *let $y$ be a target variable and $\hat{y}$ a model of $y$. For any $i$ and $j$, we define the binary target variable $y_i$ which is the indicator of event $\{y = i\}$, and the binary model $\hat{y}_j$ of $y_i$ which is the indicator of event $\{\hat{y} = j\}$. When $i = j$, each model $\hat{y}_j$ of the target variable $y_j$ is referred to as a "One versus Rest" model.*

**Proposition 6.** *Using the notations of Definition 8, we have*

$$\text{Lift}(y_i = 1, \hat{y}_j = 1) = \text{Lift}(y = i, \hat{y} = j)$$

*Proof.* We want to calculate $\mathrm{Lift}(A, B)$ with $A = \{y_i = 1\}$ and $B = \{\hat{y}_j = 1\}$. We can also write $A = \{y = i\}$ and $B = \{\hat{y} = j\}$, which is enough to conclude. $\qquad\square$

We have seen with Proposition 4 and Corollary 1 that lifts can be expressed in terms of likelihood ratios and converge towards them. This is always true in the multiclass case.

**Theorem 1.** *For any $j$,*

$$\mathbb{P}(\hat{y} = j \mid y = j) - \mu_j = \sum_i \lambda_i (\mathbb{P}(\hat{y} = j \mid y = j) - \mathbb{P}(\hat{y} = j \mid y = i))$$

*Equivalently, if all $n_{i,j}$ are non-zero:*

$$\frac{1}{\mathrm{Lift}(y = j, \hat{y} = j)} = \sum_i \frac{\lambda_i}{\mathrm{LR}_{i,j}}$$

*Proof.* We have $\mathbb{P}(\hat{y} = j \mid y = j) - \mathbb{P}(\hat{y} = j) = \mathbb{P}(\hat{y} = j \mid y = j) - \sum_i \lambda_i \mathbb{P}(\hat{y} = j \mid y = i) = \sum_i \lambda_i (\mathbb{P}(\hat{y} = j \mid y = j) - \mathbb{P}(\hat{y} = j \mid y = i))$. By dividing by $\mathbb{P}(\hat{y} = j \mid y = j)$ and subtracting 1, we deduce the second formulation. $\qquad\square$

**Corollary 2.** *If all probabilities $\mathbb{P}(\hat{y} = k \mid y = l)$ are fixed, for any $i$ and $j$ we have*

$$\lim_{\lambda_i \to 1} (\mathbb{P}(\hat{y} = j \mid y = j) - \mu_j) = \mathbb{P}(\hat{y} = j \mid y = j) - \mathbb{P}(\hat{y} = j \mid y = i)$$

*Equivalently, if all $\mathbb{P}(\hat{y} = k \mid y = l)$ are non-zero:*

$$\lim_{\lambda_i \to 1} \mathrm{Lift}(y = j, \hat{y} = j) = \mathrm{LR}_{i,j}$$

In the following, we sometimes write $\mathrm{Lift}(y = i, \hat{y} = j) \geq 1$ or $\mathrm{LR}_{i,j} \geq 1$ because these expressions are simple, but they are not always well defined. The reader can mentally replace them with $P(\hat{y} = j \mid y = i) \geq \mu_j$ or $\mathbb{P}(\hat{y} = j \mid y = j) \geq \mathbb{P}(\hat{y} = j \mid y = i)$ which are equivalent and always defined.

Intuitively, if a multiclass model is decent, we want the events $\{y = i\}$ and $\{\hat{y} = i\}$ to attract each other. In other words we want to have $\mathrm{Lift}(y = i, \hat{y} = i) \geq 1$ for any $i$, which is equivalent to saying that all One versus Rest binary models are decent according to Propositions 2 and 6. And being decent must be a characteristic of the model, independent of the $\lambda_i$ values. Corollary 2 then implies that the likelihood ratios are greater than or equal to 1. Conversely, if the likelihood ratios are greater than or equal to 1, the previous theorem shows that we have $\mathrm{Lift}(y = i, \hat{y} = i) \geq 1$ for any $i$, which leads us to the next definition. It does not involve the vector of prevalences $(\lambda_i)_i$, so it is indeed a characteristic of the model, independent of any oversampling or undersampling of the data set under consideration.

**Definition 9** (Decent multiclass model). *A model $\hat{y}$ of $y$ is decent if the following conditions are met:*

- *For any $j$, we have $\max_i \mathbb{P}(\hat{y} = j \mid y = i) = \mathbb{P}(\hat{y} = j \mid y = j)$.*

- *There are $i$ and $j$ distinct such that $\mathbb{P}(\hat{y} = j \mid y = i) < \mathbb{P}(\hat{y} = j \mid y = j)$.*

A model that satisfies the first condition but not the second is uninformative, since we then have $\mathbb{P}(\{\hat{y} = j\} \cap \{y = i\}) = \mathbb{P}(y = i)\mathbb{P}(\hat{y} = j \mid y = i) = \mathbb{P}(y = i)\mathbb{P}(\hat{y} = j \mid y = j) = \alpha_i \beta_j$ for any $i$ and $j$, using the notations from a previous remark. We can rewrite the first condition in a more familiar way, noting $x$ the observed value of $\hat{y}$, $\theta$ the value to be estimated of $y$ and $\hat{\theta}(x)$ the set $argmax_\theta \mathbb{P}(x \mid \theta)$: for any $x$ we have $x \in \hat{\theta}(x)$. We deduce that *a model is decent or uninformative if and only if it is a maximum likelihood estimator of the target variable.* When $m = 2$ we obtain that a model is decent or uninformative if we have specificity $\geq 1 - $ sensitivity and sensitivity $\geq 1 - $ specificity, that is $J \geq 0$. In the multiclass situation, several equivalences become implications: a model is decent or uninformative if and only if $\mathrm{LR}_{i,j} \geq 1$ for

any $i \neq j$, which implies $\mathrm{DOR}_{i,j} \geq 1$. According to Theorem 1, this also leads to $\mathrm{Lift}(y = j, \hat{y} = j) \geq 1$ for any $j$ and one of these inequalities is strict if the model is decent. Applying Lemma 1, we finally have $\mathrm{Lift}(y \neq j, \hat{y} = j) \leq 1$ and $\mathrm{Lift}(y = j, \hat{y} \neq j) \leq 1$ for any $j$.

Models that are neither decent nor uninformative cannot be called erroneous as in the two-class tasks: a simple permutation of the confusion matrix columns is not enough to make them decent in general, as we shall see a little further on. In other words, permuting the values predicted by the model does not correct the model's shortcomings. Any model that is not decent or uninformative in the multiclass setting is simply referred to as a "bad model". Being a decent multiclass model is a rather demanding condition: if there is a class $k$ such that $\mu_k > 0$ and $n_{k,k} = 0$, the model is bad. Poor model performance on one of the classes cannot therefore be compensated for by performance on the other classes.

The following two examples show that accuracy is a misleading metric when $m > 2$, even if we restrict ourselves to balanced data sets: it is not sufficient on its own to determine the quality of a model.

**Example 2.** *The confusion matrix* $B = \begin{pmatrix} 0 & 3 & 0 \\ 1 & 2 & 0 \\ 0 & 0 & 3 \end{pmatrix}$ *is that of a bad model according to the previous remark. However, 5 of the 9 observations are correctly classified. And we noticed after Proposition 5 that on a balanced data set with $m = 3$, an uninformative model correctly classifies only a third of the observations.*

**Example 3.** *Let* $D = \begin{pmatrix} 11 & 10 & 9 \\ 10 & 10 & 10 \\ 9 & 9 & 12 \end{pmatrix}$ *and* $U = \begin{pmatrix} 10 & 10 & 10 \\ 10 & 10 & 10 \\ 10 & 10 & 10 \end{pmatrix}$. *The confusion matrix $D$ is of rank 3, so the model is not uninformative, and we have $n_{i.} = 30$ for any $i$. When the sample is balanced, as it is here, verifying that a model is decent or uninformative is easy: we need only check that each diagonal entry is equal to the maximum value of its column, which is clear. The percentage of well-classified observations is $\frac{33}{90} = 36,66...\%$ which may seem low, and we'll come back to this rate in the next section. But our model is clearly better than the uninformative model with confusion matrix $U$.*

The model studied here looks good on one data set and bad on the oversampled data set, which shows that examining binary One versus Rest models is not enough to judge the quality of a model either.

**Example 4.** *Let* $B_1 = \begin{pmatrix} 2 & 1 & 2 \\ 3 & 2 & 0 \\ 0 & 1 & 4 \end{pmatrix}$ *and* $B_2 = \begin{pmatrix} 2 & 1 & 2 \\ 9 & 6 & 0 \\ 0 & 1 & 4 \end{pmatrix}$. *The confusion matrix $B_1$ is defined on a balanced data set, and the model is clearly bad. However, if we place the three binary One versus Rest models in the ROC space, they are all above the line of uninformative models. The confusion matrix $B_2$ is obtained by oversampling the class "1", whose numbers have been tripled, and the "0" versus Rest associated model has the confusion matrix $\begin{pmatrix} 6+0+1+4 & 9+0 \\ 1+2 & 2 \end{pmatrix}$. This model is slightly below the random guessing line of the ROC space, as $(1 - \mathrm{specificity}, \mathrm{sensitivity}) = (\frac{9}{20}, \frac{2}{5}) = (0.45, 0.4)$. Note that an over- or undersampling that leads to such an apparently contradictory situation is impossible if $m = 2$. What the two confusion matrices have in common is the value of the likelihood ratio $\mathrm{LR}_{1,0} = \frac{n_{0,0}}{n_{1,0}} \frac{\lambda_1}{\lambda_0} = \frac{2}{3} < 1$.*

*We should also point out that the maximum value of each column of $B_1$ is never in the first row. So no permutation of the columns can transform this bad model into a decent one.*

We now study a confusion matrix with a zero determinant, but corresponding to a decent model. For any confusion matrix, we have lifts greater than or equal to 1 and less than or equal to 1 in each row and each column: for any $i$ we have $\sum_j \mathbb{P}(\{y = i\} \cap \{\hat{y} = j\}) = \lambda_i = \sum_j \lambda_i \mu_j$, that is $0 = \sum_j (\mathbb{P}(\{y = i\} \cap \{\hat{y} = j\}) - \lambda_i \mu_j)$, and similarly in each column. In the case of a decent model, one might think that the off-diagonal lifts are less than 1, but the following example shows that this is not true.

**Example 5.** *We denote* $D = \begin{pmatrix} 2 & 1 & 1 \\ 1 & 2 & 1 \\ 1 & 2 & 1 \end{pmatrix}$ *and* $L = \begin{pmatrix} 1.5 & 0.6 & 1 \\ 0.75 & 1.2 & 1 \\ 0.75 & 1.2 & 1 \end{pmatrix}$. *The confusion matrix $D$ is of rank 2 (the sum of the first two columns is proportional to the third column), it is therefore not that of a uninformative model and its determinant is zero. The data set is balanced and the model is clearly decent. We have*

$\lambda_0 = \lambda_1 = \lambda_2 = \frac{1}{3}$, $\mu_0 = \frac{1}{3}$, $\mu_1 = \frac{5}{12}$, $\mu_2 = \frac{1}{4}$ *and* $(\text{Lift}(y = i, \hat{y} = j))_{i,j} = L$. *Note in particular that the off-diagonal lift* $\text{Lift}(y = 2, \hat{y} = 1)$ *is greater than 1.*

If a model is decent, Proposition 6 shows that merging all but one of the classes results in a decent One versus Rest model. But this is not always the case if the mergers are partial.

**Example 6.** *Let* $D = \begin{pmatrix} 3 & 1 & 2 & 2 \\ 2 & 2 & 2 & 2 \\ 2 & 2 & 2 & 2 \\ 3 & 2 & 1 & 2 \end{pmatrix}$. *The confusion matrix D corresponds to a decent model on a balanced data set. If we merge classes 0 and 1 and classes 2 and 3, we obtain the confusion matrix* $\begin{pmatrix} 8 & 8 \\ 9 & 7 \end{pmatrix}$ *of a bad model.*

We could try to imitate the definition of a decent model to compare two models: say that a decent model $\hat{y}_1$ of the target variable $y$ is better than a decent model $\hat{y}_2$ if all the diagonal lifts of $\hat{y}_1$ are greater than those of $\hat{y}_2$, whatever the prevalences. This would be equivalent to saying that all likelihood ratios of $\hat{y}_1$ are greater than those of $\hat{y}_2$. But already when $m = 2$ this path leads nowhere: consider a convex ROC curve (we know that we can always reduce ourselves to this situation). Let $A$ be the point associated with model $\hat{y}_1$ and $B$ with model $\hat{y}_2$, and let O, P, Q, R be the points with coordinates (0,0), (1,0), (1,1) and (0,1) respectively. We have $\text{LR}_+(\hat{y}_1) > \text{LR}_+(\hat{y}_2)$ if and only if $\widehat{AOP} > \widehat{BOP}$, and $\text{LR}_-(\hat{y}_1) < \text{LR}_-(\hat{y}_2)$ if and only if $\widehat{AQR} < \widehat{BQR}$. This is equivalent to saying that $\widehat{AOQ} > \widehat{BOQ}$ and $\widehat{AQO} > \widehat{BQO}$. We leave it to the reader to convince himself that this is impossible due to convexity, unless A=B.

## 4.2 Global Metrics

To compare several models of the same target variable, we keep the decent models and estimate their respective overall performance with a metric. Multiclass versions of Matthews' correlation coefficient or Cohen's $\kappa$ can be used, for example.

**Definition 10.**

$$\text{MCC} = \frac{\sum_i (\mathbb{P}(y = \hat{y} = i) - \lambda_i \mu_i)}{\sqrt{1 - \sum_i \lambda_i^2} \sqrt{1 - \sum_i \mu_i^2}}$$

$$\kappa = \frac{\sum_i (\mathbb{P}(y = \hat{y} = i) - \lambda_i \mu_i)}{1 - \sum_i \lambda_i \mu_i}$$

The denominator of $\kappa$ never cancels, and that of MCC cancels only for constant models. Theorem 1 shows that the numerator of these two metrics is positive if the model is decent. We have seen that lifts minus 1 are interpreted with One versus Rest models of Definition 8, so MCC and $\kappa$ are forms of macro-average.

We can extend Youden's $J$ to the multiclass case. If we still want $J$ to detect decent but not very informative models, we can set $J = \frac{1}{m-1} \left( \sum_i \mathbb{P}(\hat{y} = i \mid y = i) - 1 \right) = \frac{1}{m-1} \sum_i (\mathbb{P}(\hat{y} = i \mid y = i) - \mu_i)$. As $\hat{y}$ is decent or uninformative, $J$ lies between 0 and 1 and all terms of the last sum are non-negative, so $\hat{y}$ is close to an uninformative model if and only if all terms are small, that is if $J$ is small. In this way, we've generalized all the binary results of the Propositions 2 and 3, replacing certain equivalences with implications. The last formula can be rewritten as $J = \frac{1}{m-1} \sum_i \frac{1}{\lambda_i} (\mathbb{P}(y = \hat{y} = i) - \lambda_i \mu_i)$. When all $\lambda_i$ are equal (and therefore equal to $\frac{1}{m}$), we obtain $\kappa = J = \frac{m}{m-1} (\sum_i \mathbb{P}(y = \hat{y} = i) - \frac{1}{m})$. Thus decent multiclass models have a rate of well-classified observations greater than $\frac{1}{m}$ on balanced data sets, but we saw in Example 2 that the converse is not true when $m > 2$. Multiclass balanced accuracy is easy to define: let balanced accuracy $= \frac{1}{m} \sum_i \mathbb{P}(\hat{y} = i \mid y = i)$, which is $\lambda$-invariant. On a balanced data set, this is the rate of well-classified observations. On any data set, we have $J = \frac{1}{m-1} (m \text{ balanced accuracy} - 1)$, and so decent multiclass models always have a balanced accuracy greater than $\frac{1}{m}$. It should be noted that our extension of Youden's $J$ differs from that of Powers (2011), which is given by the formula $B(R, P) = \sum_i Prev(l)B(l) = \sum_i \lambda_i \frac{(a(i,i)d(i,i) - b(i,i)c(i,i))/n^2}{\lambda_i(1-\lambda_i)} = \sum_i \frac{\lambda_i \mu_i (\text{Lift}(y = \hat{y} = i) - 1)}{1 - \lambda_i} = \sum_i \frac{\lambda_i}{1 - \lambda_i} (\mathbb{P}(\hat{y} = i \mid y = i) - \mu_i)$. But this formula doesn't detect almost uninformative models. We summarize these considerations in the following definition.

**Definition 11.**

$$\text{Balanced accuracy} = \frac{1}{m} \sum_i \mathbb{P}(\hat{y} = i \mid y = i)$$

$$\text{J} = \frac{1}{m-1}(m \text{ balanced accuracy} - 1) = \frac{1}{m-1} \sum_i \frac{\mathbb{P}(y = \hat{y} = i) - \lambda_i \mu_i}{\lambda_i}$$

### 4.3 Boundary between Decent and Bad Models in the ROC Space

When $m = 2$, uninformative models separate decent and bad models in the ROC space. We show that this is no longer the case if $m \geq 3$ by describing precisely the boundary between these two types of models. The ROC space is defined below as a subset of $[0, 1]^{m^2}$ with the usual Euclidean topology. And for any subset $B$ of $\mathbb{R}^{m^2}$, its boundary is $\overline{B} \cap \overline{B^c}$ where $\overline{B}$ denotes the closure of $B$.

**Definition 12.** *The set of row stochastic matrices, that is square matrices with $m$ rows and $m$ columns, with non-negative coefficients and whose sum of each row is 1, is called ROC space. For any model $\hat{y}$ of a target variable $y$, we call "normalized confusion matrix" the matrix of conditional probabilities $(\mathbb{P}(\hat{y} = j \mid y = i))_{i,j}$, which is an element of the ROC space. By abuse of terminology, we refer to a decent (resp. uninformative, bad) model in the ROC space to designate the normalized confusion matrix associated with such a model.*

A first geometric remark can be made about decent or uninformative models: they form a convex subset of the ROC space, as can be seen immediately from the first condition of Definition 9.

Edwards et al. (2005) show that the hypervolumes under the ROC hypersurface of a perfect model and a random classifier are equal to 0 when $m \geq 3$ in a particular coordinate system: the diagonal elements of the matrix $(\mathbb{P}(\hat{y} = j \mid y = i))_{i,j}$ are linear combinations of the off-diagonal entries, so they use the $m(m-1)$ values $(\mathbb{P}(\hat{y} = j \mid y = i))_{i,j}$ with $i$ different from $j$ as the coordinate system in ROC space (with constraints: the sum of off-diagonal entries in each row is less than or equal to 1). When $m = 2$, these coordinates are $(1 - \text{specificity}, 1 - \text{sensitivity})$. According to the authors, "this suggests that hypervolume may not be a useful performance metric in N-class classification task for N > 2". This conclusion needs to be qualified: in the proposition that follow, we show that uninformative models are only a tiny subset of the topological boundary between decent models and bad models when $m \geq 3$. It is therefore more appropriate to compare the volumes occupied by these two categories of models, which we do in the next section when $m = 3$.

We can see from Example 3 that it's easy to distort an uninformative model a little into a decent or bad model, so these models are a subset of the boundary between decent and bad models. And they form a low-dimensional set: according to Proposition 5, the normalized confusion matrices of these models have all their rows equal. The example below shows that the boundary considered is strictly larger.

**Example 7.** *Let $C_t = \frac{1}{4} \begin{pmatrix} 2t & 3-2t & 1 \\ 1 & 2 & 1 \\ 1 & 1 & 2 \end{pmatrix}$ be a normalized confusion matrix with $0 \leq t \leq 1$. It is never the matrix of an uninformative model, since the last two rows are not proportional. The matrix $C_t$ is that of a decent model for any $t \geq 0.5$ and that of a bad model otherwise. So $C_{0.5}$ is a boundary point of the set of decent or uninformative models in the ROC space, and it's not an uninformative model. Note also that we have $\kappa = J = \frac{t}{4} \geq 0$. This shows that in the multiclass case, a classical metric can be zero or positive even when the model is bad.*

The proposition below formalizes the observations made in the previous example.

**Definition 13.** *In the ROC space, we denote $E$ the set of decent or uninformative models, that is $E = \{(M_{i,j})_{i,j} \mid \forall j, \max_i M_{i,j} = M_{j,j}\}$. And for any $k$ and $l \neq k$, let $E_{k,l} = \{(M_{i,j})_{i,j} \in E \mid M_{k,l} = M_{l,l}\}$.*

**Proposition 7.** *The boundary of $E$ is $\bigcup_{k \neq l} E_{k,l}$ and the set of uninformative models is $\bigcap_{k \neq l} E_{k,l}$.*

*Proof.* Let $M$ be an element of the boundary of the set of decent or non-informative models. This means that there is a sequence $(D(n))_n$ of decent or uninformative models such that $\lim_{n \to \infty} D(n) = M$ and a sequence $(B(n))_n$ of bad models such that $\lim_{n \to \infty} B(n) = M$. Denoting $(D(n)_{i,j})_{i,j}$ the normalized confusion matrix

of $D(n)$, we have the equations $\max_i D(n)_{i,j} = D(n)_{j,j}$ for any $j$. By letting $n$ tend to infinity, we deduce that $M$ is decent or uninformative.

Moreover, for any $n$, there exist $i(n)$ and $j(n) \neq i(n)$ such that $B(n)_{i(n),j(n)} > B(n)_{j(n),j(n)}$. There are at most $m(m-1)$ distinct pairs $(i(n),j(n))$, there is therefore at least one pair $(I,J)$ for which there exists an infinite subsequence of $(i(n),j(n))_n$ taking this value $(I,J)$. We replace the sequence $(i(n),j(n))_n$ by this subsequence, giving $B(n)_{I,J} > B(n)_{J,J}$ for any $n$. When $n$ goes to infinity, we get $M_{I,J} \geq M_{J,J}$ and since $M$ is decent or uninformative, we actually have $M_{I,J} = M_{J,J}$ and $M \in E_{I,J}$.

Conversely, if we have $M \in E_{I,J}$, for any integer $n \geq 1$ we define a matrix $(B(n))_n$ as follows:

- If $M_{I,J} = M_{J,J} = 1$ (and therefore $M_{J,j} = 0$ for any $j \neq J$), we pose $B_{J,J} = M_{J,J} - \frac{1}{n}$, $B_{J,I} = \frac{1}{n}$ and $B_{k,l} = M_{k,l}$ otherwise. In particular, we have $B_{I,J} = M_{I,J} = 1 > B_{J,J}$.

- If $M_{I,J} = M_{J,J} < 1$, there exists $k \neq J$ with $M_{I,k} > 0$. Let $B_{J,J} = M_{J,J}$, $B_{I,J} = M_{I,J} + \frac{1}{n} > M_{I,J} = M_{J,J} = B_{J,J}$, $B_{I,k} = M_{I,k} - \frac{1}{n}$ and $B_{k,l} = M_{k,l}$ otherwise. For $n$ large enough we have $B_{I,J} \leq 1$ and $B_{I,k} \geq 0$.

We thus obtain a sequence of row stochastic matrices corresponding to bad models, such that $\lim_{n \to \infty} B(n) = M$. Moreover, by definition, the point $M$ is an element of $E$, so it's a boundary point of $E$. Finally, the statement on $\bigcap_{k \neq l} E_{k,l}$ is clear. $\qquad\square$

In the binary case, we have $E_{0,1} = E_{1,0}$ and the boundary between decent and bad models is equal to the set of uninformative models.

### 4.4 Volume Under the ROC Surface for Three Classes

In this section, we allow ourselves an abuse of language: to follow the notations and terminology of Ferri et al. (2003), we speak of volume under the ROC surface. This is in fact the volume occupied by bad models in the ROC space, we never consider a particular ROC surface associated with a predictive score.

To our knowledge, there is only one article in the literature that mentions necessary and sufficient conditions for a multiclass model to be decent. In Ferri et al. (2003), we find a characterization of bad models for $m = 3$, noting $(\mathbb{P}(\hat{y} = j \mid y = i))_{i,j} = \begin{pmatrix} y_a & x_3 & x_5 \\ x_1 & y_b & x_6 \\ x_2 & x_4 & y_c \end{pmatrix}$: there exists $h_a \geq 0$, $h_b \geq 0$, $h_c \geq 0$ with $h_a + h_b + h_c = 1$ such that $x_1 \geq h_a$, $x_2 \geq h_a$, $x_3 \geq h_b$, $x_4 \geq h_b$, $x_5 \geq h_c$, $x_6 \geq h_c$.

Models that meet these criteria are indeed uninformative or bad models: we find $y_a = 1 - x_3 - x_5 \leq 1 - h_b - h_c = h_a$, and similarly $y_b \leq h_b$ and $y_c \leq h_c$. So $y_a \leq \min(x_1, x_2)$, $y_b \leq \min(x_3, x_4)$ and $y_c \leq \min(x_5, x_6)$. But these criteria are incorrect because they are too restrictive, as we can see with the balanced confusion matrix $\begin{pmatrix} 0 & 2 & 1 \\ 0 & 1 & 2 \\ 1 & 1 & 1 \end{pmatrix}$. The proportion of correctly classified observations is equal to $\frac{2}{9} < \frac{1}{m}$, so we have $J < 0$ and this model is neither decent nor uninformative. The reader can easily check that all diagonal lifts are strictly less than 1 (in other words, all three One versus Rests models are wrong), which also implies that MCC and $\kappa$ are negative. But we have $r_1 = \min(x_1, x_2) = \min(0, \frac{1}{3}) = 0$, $r_2 = \min(x_3, x_4) = \min(\frac{2}{3}, \frac{1}{3}) = \frac{1}{3}$, $r_3 = \min(x_5, x_6) = \min(\frac{1}{3}, \frac{2}{3}) = \frac{1}{3}$ and $r_1 + r_2 + r_3 = \frac{2}{3} < 1$. It is therefore not possible to find a triplet $(h_a, h_b, h_c)$ that satisfies the above conditions.

When $m = 2$, the maximum (resp. minimum) area under a ROC curve is that of a perfect probabilistic (resp. random) model, and is 1 (resp. 0.5). For the three-class case, their analogues are $\text{VUS}_3^{max}$ and $\text{VUS}_3^{min}$, noting VUS as the Volume Under the ROC Surface: $\text{VUS}_3^{max}$ is the volume of the set of all models in the ROC space and $\text{VUS}_3^{min}$ is the volume of the set of all bad models. The maximum volume calculated in Ferri et al. (2003) is $\text{VUS}_3^{max} = \frac{1}{8} = 0.125$ . We also find a Monte-Carlo estimate of $\text{VUS}_3^{min} \simeq 0.0055$, but the latter value is too low because the number of bad models is underestimated. We obtain a new Monte-Carlo

estimate for 100,000,000 points: $\mathrm{VUS}_3^{min} \simeq 0.1125$. Its exact value is calculated below, and shows that bad models occupy 90% of the space of all three-class models.

**Proposition 8.**

$$\mathrm{VUS}_3^{min} = \frac{9}{80} = 0.9\,\mathrm{VUS}_3^{max}$$

*Proof.* Consider the normalized confusion matrix $X = \begin{pmatrix} x_{00} & x_{01} & x_{02} \\ x_{10} & x_{11} & x_{12} \\ x_{20} & x_{21} & x_{22} \end{pmatrix}$ subject to the following conditions: $0 \leq x_{ij} \leq 1$ for any $i$ and $j$ and $\sum_j x_{ij} = 1$ for any $i$. In the volume calculations below, we can impose that the $x_{ij}$ are all distinct. All these matrices occupy a hypersurface of dimension 6 of volume $\mathrm{VUS}_3^{max}$. We want to calculate the volume $\mathrm{VUS}_3^{decent}$ of matrices $X$ where the maximum value of each column $j$ is reached in row $j$. By symmetry, we have $V = 6\,\mathrm{VUS}_3^{decent}$ where $V$ is the volume of matrices $X$ where the maximum values of each column are reached in distinct rows. So $W = \mathrm{VUS}_3^{max} - V$ is the volume of matrices $X$ where the maximum values of each column are reached in two of the three rows. Indeed, we can't have the three maximum values of each column in a single row, because of the constraints $\sum_j x_{ij} = 1$. Again by symmetry, we have $W = 9\,U$ with $U$ the volume of matrices $X$ where the maximum values of the first two columns are reached in the first row. Finally, we have $\mathrm{VUS}_3^{min} = \mathrm{VUS}_3^{max} - \mathrm{VUS}_3^{decent} = \frac{5}{6}\mathrm{VUS}_3^{max} + \frac{3}{2}U$. All that remains is to calculate $U$, the constraints being $0 \leq x_{10} < x_{00}$, $0 \leq x_{20} < x_{00}$, $0 \leq x_{11} < x_{01}$, $0 \leq x_{21} < x_{01}$, $x_{00} + x_{01} \leq 1$, $x_{10} + x_{11} \leq 1$ and $x_{20} + x_{21} \leq 1$. The last two inequalities are consequences of the other inequalities, which gives

$$\begin{aligned}
U &= \int_0^1 dx_{00} \int_0^{1-x_{00}} dx_{01} \int_0^{x_{00}} dx_{10} \int_0^{x_{00}} dx_{20} \int_0^{x_{01}} dx_{11} \int_0^{x_{01}} dx_{21} \\
&= \int_0^1 x_{00}^2\, dx_{00} \int_0^{1-x_{00}} x_{01}^2\, dx_{01} = \int_0^1 x^2\, dx \int_0^{1-x} y^2\, dy = \frac{1}{3} \int_0^1 x^2 (1-x)^3\, dx \\
&= \frac{1}{3} \int_0^1 (x^2 - 3x^3 + 3x^4 - x^5)\, dx = \frac{1}{3}\left(\frac{1}{3} - \frac{3}{4} + \frac{3}{5} - \frac{1}{6}\right) = \frac{1}{3}\left(\frac{1}{6} - \frac{3}{20}\right) = \frac{1}{180}
\end{aligned}$$

We deduce $\mathrm{VUS}_3^{min} = \frac{5}{6}\frac{1}{8} + \frac{3}{2}\frac{1}{180} = \frac{1}{12}\frac{5}{4} + \frac{1}{12}\frac{1}{10} = \frac{1}{12}\frac{27}{20} = \frac{9}{80}$. $\qquad\square$

There are several possible extensions of the ROC curve to the multiclass setting and other definitions of ROC space, VUS and multiclass AUC, which may be more useful in practice (see Kleiman & Page 2019 or Aguilar-Ruiz & Michalak 2024 for example). We won't discuss these variants, as they stray too far from the subject of our article.

### 4.5 Weak Multiclass Classifiers

In this final section, we show how inequalities on likelihood ratios can be used to define weak multiclass classifiers. We state the next proposition using balanced accuracy, which is a linearly transformed version of Youden's $J$ according to Definition 11 (but in no case do we assume that the data sets considered are balanced). We have already pointed out in the discussion following Definition 10 that if we restrict ourselves to decent models, a value of $J$ close to 0, that is a balanced accuracy close to $\frac{1}{m}$, means that the model is not very informative. And if the balanced accuracy is is nearly equal to its maximum value 1, then the normalized confusion matrix is very close to the diagonal identity matrix and the model is almost perfect.

**Proposition 9.** *Let $\delta > 0$ and $\gamma > 1$ be such that $1 + \delta \leq \mathrm{LR}_{i,j} \leq \gamma$ for any distinct $i$ and $j$. Then*

$$\frac{1}{m} + \delta\frac{m-1}{m(m+\delta)} = \frac{1+\delta}{m+\delta} \leq \text{balanced accuracy} \leq \frac{\gamma}{m}$$

*Proof.* We have $\mathrm{LR}_{j,j} = 1$, so $\mathrm{LR}_{i,j} \leq \gamma$ for any $i$ and $j$. Theorem 1 leads to $\mathrm{Lift}(y = j, \hat{y} = j) \leq \gamma$, that is $\mathbb{P}(\hat{y} = j \mid y = j) \leq \gamma\mu_j$ for any $j$. We deduce balanced accuracy $\leq \frac{\gamma}{m}$ by summing over $j$.

For $i$ fixed and any $j \neq i$, we have $\mathbb{P}(\hat{y} = j \mid y = j) \geq (1 + \delta)\mathbb{P}(\hat{y} = j \mid y = i)$, and we also have $\mathbb{P}(\hat{y} = i \mid y = i) = (1+\delta)\mathbb{P}(\hat{y} = i \mid y = i) - \delta\mathbb{P}(\hat{y} = i \mid y = i)$. Summing over $j$, we get $m$ balanced accuracy $\geq 1 + \delta - \delta\mathbb{P}(\hat{y} = i \mid y = i)$. Then we sum over $i$, which gives $m^2$ balanced accuracy $\geq m(1 + \delta) - \delta m$ balanced accuracy. Dividing by $m$ we finally find $(m + \delta)$ balanced accuracy $\geq 1 + \delta$. $\qquad\square$

The assumptions in the previous proposition guarantee that the model does better than chance, and they allow us to control how close or how far the balanced accuracy of the model is to that of a random model. Such inequalities can therefore be a good starting point for defining weak multiclass classifiers used, for example, by a boosting algorithm. These inequalities could also remove a theoretical barrier: in Schapire & Freund (2012), we find an example of two weak classifiers $h_1 = \begin{pmatrix} 1 & 0 & 0 \\ 1 & 0 & 0 \\ 0 & 0 & 1 \end{pmatrix}$ and $h_2 = \begin{pmatrix} 0 & 1 & 0 \\ 0 & 1 & 0 \\ 0 & 0 & 1 \end{pmatrix}$ "with accuracy significantly better than the random guessing rate of $\frac{1}{m}$, but for which no boosting algorithm can exist that uses such weak classifiers to compute a combined classifier with perfect (training) accuracy". The authors add that "This difficulty turns out to be provably unavoidable when the performance of the weak learner is measured only in terms of error rate". This led them to completely change their approach, abandoning weak multiclass classifiers and reducing the multiclass problem to several binary problems. We immediately check that $\mathrm{LR}_{1,0}(h_1) = \mathrm{LR}_{0,1}(h_2) = 1$. So this counter-example doesn't apply if we no longer use accuracy but rather likelihood ratios as in the previous proposition to define the weakness of a classifier. It may then be possible to design a multiclass boosting algorithm as close to perfection as desired on the training sample.

## 5   Conclusion

We obtain a characterization of decent multiclass models based on two assumptions: being decent is a characteristic of the model, independent of the frequency of each class of the target variable. And a decent model must have decent One versus Rest binary versions. This gives us some very simple necessary and sufficient conditions, which generalize in a natural way various well-known criteria for two-class problems. These criteria have enabled us to correct or relativize several previous results, and they also pave the way for the design of new algorithms for aggregating multiclass weak classifiers.

Decent models are more complex to understand intuitively when $m > 2$. Even if we restrict ourselves to balanced data sets, the rate of correct classification no longer tells us whether a model is decent or not. Merging certain classes can transform a decent model into a bad one, unless we apply a One versus Rest merger. Uninformative models no longer form the boundary between decent and bad models. And there's no natural way to generalize binary metrics into global multiclass metrics, unless you want to preserve particular properties of the metric.

Unexpectedly, we show that the classical metrics seen in the binary case actually hide two groups of metrics. We have families of pointwise metrics, associated with each cell of the confusion matrix: lifts, diagnostic odds ratios and likelihood ratios. The latter are used to define decent multiclass models. We also have global metrics such as Matthews' correlation coefficient, Cohen's $\kappa$ and Youden's J., they take on positive values on decent models.

### Acknowledgments

I'd like to thank my colleagues from the Customer Knowledge team who attend my Python training courses. This forces me to go into greater depth on certain subjects to ensure that they are as accurate and complete as possible. I'd also like to thank Célia Etien for her patient proofreading and valuable feedback.

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
