# OpenReview forum: "Mathematical Characterization of Better-than-Random Multiclass Models"
_TMLR — Accepted by TMLR_

### Review · Reviewer_wubw · 2025-04-05

**Summary Of Contributions:**

The authors extend the existing results on supervised models outperforming chance to the multiclass setting, and discover many interesting properties.

**Audience:**

Yes

**Broader Impact Concerns:**

No ethical concerns.

**Claims And Evidence:**

Yes

**Requested Changes:**

It is odd to refer to sections of the paper as chapters. Perhaps this work is extracted from a thesis. Please consider changing "chapter" to "section".

$\lambda > 0$ and  $\lambda <1$ hold when $c+d <n$, right? That is, when at least one between $a$ and $b$ are positive.

The writing can be polished in some parts. For example, in Proposition 4 it would be customary to write something like "The following hold" before presenting the formulae. In addition, sometimes Cohen's $\kappa$ is referred to as Cohen's kappa (this is indeed very minor).

I must confess I don't quite understand the need right after Definition 9 to introduce new notation to claim that a model is decent or uninformative if and only if it is a maximum likelihood estimator of the target variable. Was it to better relate to the most commonly used notation $P(x \mid \theta)$ for the likelihood distribution?

Please do not use contraction as "let's".

**Strengths And Weaknesses:**

I must admit I'm not an expert in the subject matter, but I really enjoyed reading this paper. Given my non-expert nature, I may not be aware of other similar existing results. That being said, the propositions and theorems that the authors present are very interesting, and to some extent even surprising (I'm referring especially to the boundary in the multiclass setting being a superset of the collection of uninformative models). I'm happy to recommend for an acceptance after minor revisions, that are spelled out in the next text box.

---

### Review · Reviewer_KkCr · 2025-04-18

**Summary Of Contributions:**

The paper fully characterizes when multi-class models truly outperform chance (i.e., a classifier decision made by random coin flipping) via algebraic conditions on the confusion matrix. This extends core binary metrics, maps out the ROC‐space geometry, and even lays theoretical groundwork for multi-class scenarios.

**Audience:**

Yes

**Broader Impact Concerns:**

Not applicable.

**Claims And Evidence:**

Yes

**Requested Changes:**

Please see the comments above, all these comments should be carefully addressed.

**Strengths And Weaknesses:**

Comments:

The authors may want to define "ROC curve" and " ROC surfaces" explicitly. It is a bit odd that the authors emphasize elementary properties like Proposition 2 in details, while assuming that the readers are all familiar with ROC curves and surfaces, which are considered to be advanced concepts.

In addition, it is not rigorous to use "ROC space" and "ROC hypersurfaces" interchangeably, since "ROC space" (Clémençon et.al, 2008, AoS) can also refer to the collection of ROC curves associated with a family of classifiers in formal research. Clémençon and Robbiano (2015, JNS) have established similar results for ROC curves from the perspective of ranking. It will be helpful if the authors can comment how this result shed light on the understanding of ROC curves in the classification scenario.

Section 1: It is friendly to introduce "One versus Rest" when moving from binary to multi-class classification context, in particular, the elementary examples in Section 2 can be removed, and the room could be used to have detailed examples in Section 4, where the novelty lives. For the prerequisite concepts that are not used, we can refer the readers to relevant literature instead of listing all concepts. This revision can also help reader to understand claims "If a model is decent, merging all but one of the classes results in a decent One versus Rest model." (P11) later, which itself requires a proof.

Example 2: The phrase “rate of well classified is nonetheless not negligible” is confusing. Is the author attempting to say: “rate of correctly classified observations” or “well‑classified rate”?

Section 2.3: For "Those that do worse than chance", practitioners usually just use its complement model. Namely, if the model predicts class A, then we would make the decision of choosing class B. Therefore, the complement model will be a classifier that do better than chance. Is there any reason why we shall consider them separately?

Section 3.2: It is unclear why Proposition 4 provides new insghts. As pointed out by the authors, Vu et. al(2019) has already stated similar results; It seems like that we are describing a Bayes factor for testing the hypothesis that the sample belongs to class 0 or 1, is that correct?

Section 3.3: If this section is not directly related (as stated by the authors), it can be postponed to the appendix. Or the author may want to give examples to illustrate how normalized likelihood ratios are different from the unnormalized ones, since this CO-type metrics seem to be a linear transformed version of original metrics.

Section 4.3: "And from proposition 5, the normalized confusion matrices of uninformative models have m − 1 degrees of freedom. We can use the values (P(ˆy = j | y = i))i,j with i different from j as the coordinate system in ROC space." This is quite loose a claim since Prop 5 is for binary model and this is a claim for multi-class model. I am not sure if this is correct, and the concept of "degrees of freedom" comes out from nowhere.

Example 7: This is also confusing. From Definition 12, the normalized confusion matrix needs to have legit row probabilities, and it has to be row stochastic, then what does it mean by "we can move continuously from a decent model with all its likelihood ratios greater than 1 to a bad model". If you move in the space of parameter t, then isn't the boundary just a point? if you move in the space of row stochastic matrices, it is not clear this boundary is one of those geometric boundaries in that space (perhaps need better explanation of the geometric interpretation in previous Sec 3.2-3.3).

Proposition 7: How do we define boundary here? And the meaning of ROC space requires a rigorous math definition and topology, then we can introduce the concept of boundary. Or I may not fully understand what the author meant here.

Section 4.5: The discussion seems to be focused on the balanced multi-class classification problem, this is quite different from the generality claimed in Section 1, it needs to be clarified how well these analysis can be generalized to the unbalanced scenarios.

---

> ### Comment · Reviewer_KkCr · 2025-05-04
> **modified Audience to "Yes"**
>
> Thank you for the revision, I have modified Audience to "Yes", and feeling that the readability is improved in this revision. But the paper will further benefit from adding a more recent literature review.

---

### Review · Reviewer_vfzM · 2025-04-30

**Summary Of Contributions:**

The paper provides a formal characterization of decent models—those that perform better than random guessing—for multiclass classification tasks, based on an analysis of their confusion matrices. The authors build their argument on two key principles: (1) decency should be an intrinsic property of the model, independent of class imbalance in the dataset, and (2) a decent model in a multiclass setting should also exhibit decency when evaluated using the One-vs-Rest (OvR) approach, a well-studied binary classification framework.  These principles culminate in a formal definition of decency (Definition 9). The authors also analyze the ROC space, highlighting a critical difference from the binary case: in multiclass classification, uninformative models occupy only a small region on the boundary separating decent and bad models.

**Audience:**

Yes

**Claims And Evidence:**

Yes

**Requested Changes:**

I don't have many suggestions for changes to this paper. However, in Section 4.3, I recommend that the authors provide a more systematic introduction to the ROC space in the binary case, offering the necessary background before extending the discussion to the multiclass case ($m \geq 3$).

**Strengths And Weaknesses:**

**Strengths:**

- Extending the notion of model decency from binary to multiclass classification is non-trivial, as several properties do not carry over directly. This paper addresses these challenges thoughtfully, clearly discussing the distinctions between the two settings.

- The problem tackled is relevant to practitioners, particularly those dealing with imbalanced multiclass datasets. The framework proposed offers a principled and consistent approach to evaluating model performance in such contexts.

**Weaknesses:**

- I would expect some (preliminary) experiments for the proposed aggregating multiclass weak classifiers (e.g., in boosting context) to see if there is any advantage it might offer in practice.

---

### Decision · Action_Editor_J6o8 · 2025-05-24

**Recommendation:** Accept as is

**Comment:**

This paper provides a characterization of multiclass models that perform better than chance. The results are non-trivial and could be of interest to practitioners.

**Audience:**

The problem tackled is relevant to practitioners, particularly those dealing with imbalanced multiclass datasets.

**Claims And Evidence:**

The reviewers are satisfied with the clarity of the claims and the supporting proofs.